# Split-Belt Treadmill Adaptation Improves Spatial and Temporal Gait Symmetry in People with Multiple Sclerosis

**DOI:** 10.3390/s23125456

**Published:** 2023-06-09

**Authors:** Andrew C. Hagen, Jordan S. Acosta, Chaia S. Geltser, Brett W. Fling

**Affiliations:** 1Department of Health and Exercise Science, Colorado State University, Fort Collins, CO 80523, USAbrett.fling@colostate.edu (B.W.F.); 2Department of Biomedical Sciences, Colorado State University, Fort Collins, CO 80523, USA; 3Molecular, Cellular and Integrative Neurosciences Program, Colorado State University, Fort Collins, CO 80523, USA

**Keywords:** multiple sclerosis, adaptation, split-belt treadmill, gait asymmetry, sensorimotor control, locomotion

## Abstract

Multiple sclerosis (MS) is a neurodegenerative disease characterized by degradation of the myelin sheath resulting in impaired neural communication throughout the body. As a result, most people with MS (PwMS) experience gait asymmetries between their legs leading to an increased risk of falls. Recent work indicates that split-belt treadmill adaptation, where the speed of each leg is controlled independently, can decrease gait asymmetries for other neurodegenerative impairments. The purpose of this study was to test the efficacy of split-belt treadmill training to improve gait symmetry in PwMS. In this study, 35 PwMS underwent a 10 min split-belt treadmill adaptation paradigm, with the faster paced belt moving under the more affected limb. Step length asymmetry (SLA) and phase coordination index (PCI) were the primary outcome measures used to assess spatial and temporal gait symmetries, respectively. It was predicted that participants with a worse baseline symmetry would have a greater response to split-belt treadmill adaptation. Following this adaptation paradigm, PwMS experienced aftereffects that improved gait symmetry, with a significant difference between predicted responders and nonresponders in both SLA and PCI change (*p* < 0.001). Additionally, there was no correlation between SLA and PCI change. These findings suggest that PwMS retain the ability for gait adaptation, with those most asymmetrical at baseline demonstrating the greatest improvement, and that there may be separate neural mechanisms for spatial and temporal locomotor adjustments.

## 1. Introduction

Multiple sclerosis (MS) is a neurodegenerative disease afflicting more than two million people worldwide [1] and has an average annual economic burden of USD 65,612 to each person with MS in the United States [2]. This chronic condition generally onsets between the ages of 20 and 50 and affects women three times more than men [1]. MS is characterized by the degradation of the myelin sheath, an insulating layer of lipids and proteins that increases the velocity of electrical impulse propagation along a nerve. This results in impaired neural communication to and from the brain. Typically, people with MS (PwMS) have more severe sensory and motor impairments on one side of their body [3]. This results in significant spatial and temporal gait asymmetries which subsequently leads to falls, musculoskeletal injuries, diminished engagement in daily life activities, and decreased quality of life [4]. Additionally, it has been shown that an asymmetrical gait results in a higher metabolic cost [5,6] and cognitive demand [7,8]. Accordingly, rehabilitation focused on minimizing gait asymmetries is exceedingly beneficial for PwMS to improve balance and mobility [9].

Every day, people encounter obstacles such as stairs, curbs, and different surfaces that interrupt normal gait patterns. Thus, people must adjust spatial and temporal components of their walking in order to successfully navigate these obstacles. If either of these components is not corrected for, or if a person has a pathologically derived gait asymmetry such as in PwMS [10], potentially injurious or even lethal falls may ensue.

A requisite for effective physical rehabilitation is an individual’s ability to learn and adapt. Motor adaptation relies on externally imposed perturbations from the environment that induce a trial-and-error method of adjusting movements to new demands [11,12]. One potential therapeutic intervention based on sensorimotor adaptation is a split-belt treadmill paradigm which promotes one leg to move faster than the other leg. This process creates aftereffects that alter gait symmetry via feedforward storage of a new walking pattern [13]. The importance of focusing on motor adaptation as a potential rehabilitation strategy is accentuated by the need for non-pharmacological and individualized solutions for functional motor recovery in the many populations who experience gait dysfunction [14].

Previous studies have demonstrated that people who have suffered a stroke and people with Parkinson’s disease, who experience similar sensorimotor difficulties and gait asymmetries as PwMS, are able to adapt their gait to perturbations on a split-belt treadmill [15,16,17,18]. Reisman et al. showed that repeated split-belt treadmill training improved post-stroke step length asymmetry [19], while Hulzinga et al. found that split-belt treadmill training moderately improved gait adaptation in people with Parkinson’s disease [20]. Additional studies on PwMS showed that despite diminished motor function, PwMS still have preserved motor learning abilities [21]. These alterations in gait pattern, though not permanent, are indicative of neuroplastic adaptive capabilities and provide encouraging results for potential treatment strategies. In this study, the aim was to determine if spatial and temporal gait parameters are adaptable in PwMS during split-belt treadmill adaptation and if this adaptation can lead to a decrease in gait asymmetry. We maintained a two-part hypothesis. First, it is possible to adapt and improve spatiotemporal gait symmetry in PwMS using a split-belt treadmill paradigm. Second, participants with poorer baseline symmetry, compared to averages reported in the literature for PwMS [22], would experience more significant aftereffects following split-belt treadmill adaptation.

## 2. Materials and Methods

### 2.1. Participants

We recruited a convenience sample of participants across northern Colorado. Inclusion criteria included people with relapsing-remitting MS from ages 18 to 86 that were fully ambulatory without an assistive device and could walk three-tenths of a mile without stopping to rest. This was to ensure participants could perform the split-belt treadmill adaptation paradigm safely and to avoid the effects of fatigue. Exclusion criteria included musculoskeletal injury within the previous six months, a history of brain injury, or any history of balance impairments unrelated to MS. The exclusion criteria were designed to target MS-related balance and gait impairments and avoid other confounding factors. This study was approved by the Colorado State University Biomedical Institutional Review Board (protocol code 1664).

Following screening and informed consent, demographics and patient-reported disease characteristics were collected using REDCap survey software (v. 13.1.32). Surveys collected include the Expanded Disabilities Status Scale (EDSS), Multiple Sclerosis Walking Score 12 (MSWS-12), Modified Fatigue Impact Scale (MFIS), Short Form 36 (RAND 36) Beck Depression Inventory (BDI-II), and the Montreal Cognitive Assessment (MOCA).

### 2.2. Split-Belt Treadmill Adaptation Paradigm

The participants completed five different walking trials (Figure 1). The first two trials were the overground baseline period. This consisted of two separate 2 min walk tests at their preferred walking speed and at the fastest walking speed they were comfortable. The preferred and fast walking speeds determined tied-belt and split-belt treadmill speeds. Participants completed a two-minute walk on the treadmill in the tied-belt configuration set to their preferred walk speed. Immediately following, the adaptation period began. The belts were put into split configuration with the fast belt moving at participant’s baseline fast walk speed, while the slow belt was set to half of the fast walk speed. After 10 min, the treadmill was set back to the tied-belt configuration for a 1 min post-adaptation trial at the preferred speed. 

### 2.3. Gait Analysis

Gait cycle parameters were measured during both overground and treadmill walking trials. The participants were outfitted with 6 APDM Opal inertial sensors along with 16 retroreflective markers for collection of three-dimensional motion capture data at 100 Hz. During overground walking, participants walked back and forth down a 30 m hallway, turning at each end, while APDM Mobility Lab collected inertial sensor gait cycle parameters. While treadmill walking, participants walked on a custom-built split-belt treadmill instrumented with Bertec force platforms (Model 4060-10, Bertec Corp., Columbus, OH, USA). This treadmill consisted of two separate belts, each with its own motor that permitted the speed of each belt to be controlled independently and collected ground reaction forces at 1000 Hz. The speed at which a participant walked was individualized to their overground preferred walk speed and fast walk speed. During split-belt treadmill adaptation, the fast belt speed was equal to the participant’s overground fast walk speed while the slow belt speed was half of the participants fast walk speed (2:1 ratio). Prior research determined that the fast belt should be under the more affected limb to improve gait symmetry [23]. The more affected limb was determined by participant self-reporting and investigator observation while overground walking.

### 2.4. Data Processing

Three-dimensional trajectory and force data were processed using Vicon Nexus software (v2.14, Vicon Motion Systems, Oxford, UK). Trajectory positions were filtered using a Woltring filter. Following, gait cycle events were identified from ground reaction forces using custom MATLAB software (v. 9.13.0, MathWorks Inc., Natick, MA, USA). Joint kinematics were calculated using the Vicon Plug-In Gait modeling pipeline. From a combination of trajectory and force data, gait cycle parameters and asymmetry metrics were calculated using custom MATLAB software. All gait cycle parameter means were calculated using only the second half of gait cycles from each trial to ensure gait stabilization. 

For this study, the primary outcome variable to represent spatial symmetry was step length asymmetry (SLA). Step length was calculated by taking the anterior–posterior distance between heel markers at leading limb heel strike. A body-centered model of heel location was used to mitigate the confounding effects of participant translation on the treadmill [24]. SLA was calculated by subtracting the step length of the less affected limb from the step length of the more affected limb for each consecutive step. Another spatial measure used was limb excursion asymmetry (LEA). This is a modified measure of stride length. When walking on a treadmill, participants were not translating while walking; rather, they were staying in place. Due to this, the conventional understanding of stride length (distance from heel strike to following ipsilateral heel strike) resulted in a net zero stride length. A clearer term, limb excursion, quantifies the anterior–posterior distance traveled by the limb from toe-off to ipsilateral heel strike [25]. LEA was calculated by subtracting the limb excursion of the less affected limb from the limb excursion of the more affected limb for each gait cycle.

The primary outcome variable to represent temporal symmetry was phase coordination index (PCI). PCI is a measure of stepping accuracy and consistency in relation to anti-phased stepping during walking [26]. In perfectly timed gait, each step time is exactly half of the gait cycle duration. The PCI calculation is the summation of two measures representing the relative timing of contralateral heel strikes which determines phase, represented as phi (Φ). Phi was calculated through the normalization of step time with respect to stride time of the contralateral limb (i.e., Φ = 180° for each step is ideal interlimb coordination). Once Φ was determined, the absolute error (ABS) of Φ from 180° and the covariation (CoV) of Φ were summed to give PCI, with a lower value equating to better phase coordination. These calculations provided a quantification of both absolute accuracy and relative consistency of stepping. This measure generated an index that can compare walking quality among participants, monitoring potential changes pre- and post-adaptation.
(1)$$\varphi ABS=\frac{mean\left(\left|\varphi -180{}^{\circ}\right|\right)}{180{}^{\circ}}\times 100\;\varphi CoV=\frac{stdev\left(\varphi \right)}{mean\left(\varphi \right)}\times 100\;PCI\left(\%\right)=\varphi ABS+\varphi CoV$$

### 2.5. Statistical Methods

Following baseline overground walking, participants were grouped as either responders or nonresponders. It was hypothesized that participants with a low baseline PCI or SLA (better symmetry) would experience minimal symmetry improvements following split-belt treadmill adaptation. In this analysis, participants were grouped as responders or nonresponders based on their baseline PCI value as above or below the total sample median of 5.19%, and based on their baseline SLA value as above or below the total sample median of 25.48 mm. It was predicted that participants with a baseline PCI above 5.19% or a baseline SLA above 25.48 mm would have a greater response to split-belt treadmill adaptation.

Statistical processing and figure creation were completed using R Statistical Software (v4.2.1; R Core Team, Vienna, Austria, 2022), with analysis of variance (ANOVA) calculations using the rstatix package [27]. A 2 × 2 repeated measures ANOVA was used to assess differences in PCI and SLA, with group as a between-subjects factor and timepoint (baseline versus post-adaptation) as a within-subject factor. The Bonferroni adjustment was used to account for multiple comparisons when applicable. Residuals versus fitted plots along with quantile–quantile plots were used to confirm normality, and Mauchly’s test was used to confirm sphericity. A significant two-way interaction was found between group and timepoint; therefore, simple main effect and pairwise post hoc analyses were conducted. The reported *p*-values were from pairwise comparisons. Effect sizes were calculated using Cohen’s d [28]. After the creation of a general linear model, Pearson’s product-moment correlation coefficients were calculated between changes in PCI and SLA.

## 3. Results

### 3.1. Participants

A total of 35 participants with MS completed this study with a mean age of 51.66 (12.02) and a mean of 13.85 (0.73) years since diagnosis (Table 1). This sample was quite active compared to normative PwMS [29], exercising 289.4 (266.8) minutes per week, which is typical for this geographical location. A total of 85% of participants reported symptoms of neuropathy. Additionally, this cohort scored relatively low on MS disability scales [30], with a mean EDSS of 3.57 (1.03) and mean MSWS-12 of 21.89 (12.08).

### 3.2. Spatial Symmetry

For this cohort of PwMS, there were 18 participants in the predicted responders group and 17 participants in the predicted nonresponders group, which were determined by their baseline spatial symmetry as above (responders) or below (nonresponders) an SLA of 25.48 mm. Those in the responders group experienced significant improvements in spatial symmetry (*p* = 0.050) following one session of split-belt treadmill adaptation, while the nonresponders experienced worsened symmetry (*p* < 0.001). This is represented by a change in SLA from baseline to post-adaptation, with a negative value indicating an improvement in spatial symmetry. The mean change in the responders group was −21.99 mm (SE = 12.41 mm) while the change in the nonresponders group was 38.43 mm (SE = 7.70 mm). There was also a significant difference in SLA change from baseline to post-adaptation between responders and nonresponders (*p* < 0.001) with an effect size of 1.40, as shown in Figure 2A. Along with SLA, LEA also demonstrated robust improvements in the responders group (*p* < 0.001) with a mean change of −21.29 mm (SE = 5.06 mm), compared to a mean change of 4.75 mm (SE = 2.08 mm) in the nonresponders group. A stride-by-stride analysis of SLA for a single participant (Figure 2B) showed baseline spatial asymmetry between limbs, followed by improved spatial symmetry during the post-adaptation aftereffects. 

It was hypothesized that the mechanism of spatial adaptation would be increased step length of the affected limb since the affected limb was forced to take steps on a belt moving two times as fast as the less affected limb. However, there were no observed group changes from baseline to post split-belt adaptation for step length of the affected limb when grouped together or separated into responders and nonresponders (*p* = 0.74, d = −0.11), as shown in Figure 3. This demonstrates that participants used different strategies to successfully reduce spatial asymmetry.

### 3.3. Temporal Symmetry

Temporal symmetry, represented by PCI, also significantly improved following split-belt treadmill adaptation for the responders group. There were 17 participants in the predicted responders group and 17 participants in the predicted nonresponders group, with 1 participant excluded due to equipment error. A participant was assigned to this responders group if their baseline PCI was above the total sample median PCI of 5.19% (worse temporal symmetry). It is important to note that the responders group for PCI is not the same cohort as the responders group for SLA. Those in the responders group had significant temporal symmetry improvements during the post-adaptation trial with a mean change of −1.59% (SE = 0.51%, *p* = 0.007), while those in the nonresponders group had a slight worsening of symmetry (mean = 0.71%, SE = 0.31%, *p* = 0.03). Similar to spatial symmetry outcomes, there was also a significant difference in PCI change from baseline to post-adaptation between the responders and nonresponders groups (*p* < 0.001), with an effect size of 1.33, as shown in Figure 4A. A stride-by-stride analysis of the Φ component of PCI for a single participant (Figure 4B) shows baseline temporal asymmetry between limbs with a high amount of variability, followed by improved temporal consistency during the post-adaptation trial aftereffects.

### 3.4. Correlation of Spatial Change and Temporal Change

In accordance with the second hypothesis, it was found that baseline symmetry values were predictors for symmetry improvement following adaptation, for both change in SLA (r = −0.65, *p* < 0.001) and for change in PCI (r = −0.54, *p* = 0.0012). However, individual participant spatial and temporal symmetry improvements were not correlated (r = −0.12, *p* = 0.49), as shown in Figure 5. This suggests that temporal gait parameters and spatial gait parameters may adapt independently. 

## 4. Discussion

Similar to previous work demonstrating the retention of postural motor adaptation in PwMS [31], the current study indicates that PwMS also maintain a substantial capacity for locomotor adaptation. While there were responders and nonresponders in terms of symmetry improvements, nearly all participants adapted their gait, whether this improved or worsened overall symmetry. This highlights the need for individuality in motor adaptation paradigms. Likely, each person with MS needs a different speed ratio that can be derived from their baseline symmetry, to optimize symmetry improvements on the treadmill. Additionally, baseline asymmetry predicted post-adaptation symmetry improvements both spatially and temporally. However, spatial and temporal changes were not correlated with each other, suggesting that spatial and temporal gait parameters error-correct independently. These findings are in line with split-belt treadmill training outcomes in differing neurological populations. Specifically, the maintained ability for sensorimotor adaptation [11] and the lack of correlation between spatial and temporal changes [32]. 

### 4.1. Baseline Asymmetry as a Predictor for Symmetry Improvements

It is imperative to consider the sensory system for PwMS as it has implications on motor performance. Sensory disturbances and abnormalities are observed in 80% of patients with MS [33], reducing not only neuronal signaling, but also quality of life. The importance of sensory mechanisms for balance and gait has been largely studied over a variety of populations [34]. Specifically, proprioceptive feedback is relied upon heavily for locomotor coordination and spatiotemporal planning of precise movements [35]. Ongoing research is investigating populations where sensory mechanisms are impaired and subsequently motor performance is affected [36]. Further, a growing body of literature has investigated motor adaptability despite sensory dysfunction [37]. 

Previous research utilizing a split-belt treadmill paradigm to evaluate adaptability showed improvements in both spatial and temporal asymmetries following stroke [23]. While long-term retention of sensorimotor adaptation is poorly understood, some studies found that repeated split-belt treadmill adaptation allowed for short-term adaptations and may even lead to longer-term improvements in interlimb symmetry post-stroke [19]. These findings demonstrate the capability of modulating gait cycle parameters despite cerebral damage. Therefore, this study expanded upon split-belt treadmill adaptation to PwMS, who have white matter tract damage, to further understand the mechanisms of sensorimotor adaptation and the populations it may benefit. 

These results highlight not only the adaptation potential for PwMS, but also that capacity for improvement is dependent upon baseline gait asymmetry. There was observed locomotor flexibility and reduced spatial and temporal asymmetries, providing a more coordinated gait pattern following acute split-belt treadmill adaptation. Additionally, these findings highlight that individuals with greater asymmetry at baseline experienced the largest improvements in spatial and temporal gait symmetries following split-belt adaptation. Specifically, those who had worse than average spatial and temporal asymmetries at baseline (responders) saw significantly increased improvements compared to those who were more symmetrical at baseline (nonresponders). Furthermore, this evidence illuminates locomotor adaptability in PwMS despite axonal damage and sensory deficits. Corresponding with previous findings [15], these improvements in symmetry are promising as it points to the potential of gait asymmetry being remediated with individualized adaptation paradigms. 

### 4.2. Spatial and Temporal Independence

At baseline, participants had differing levels of gait asymmetry, with some having more severe spatial deficits and some having more severe temporal deficits. The majority of PwMS experience walking difficulties and develop compensatory strategies to continue walking, even if it is inefficient. Some compensation methods include taking smaller steps with their affected limb or increasing swing time with their affected limb in order to spend more time in stance with their less affected limb. These different compensation strategies create different types of asymmetries. Along with this, some participants greatly improve their spatial symmetry following split-belt treadmill adaptation, while experiencing negligible changes in temporal symmetry and vice versa. Participants’ spatial improvements and temporal improvements were not correlated (r = −0.12, *p* = 0.49). This indicates that split-belt treadmill adaptation affects spatial and temporal gait cycle parameters independently for each participant, likely having different neural mechanisms for modulation. This finding is congruent with other studies [32] and is an important consideration when designing rehabilitation paradigms for PwMS. It is likely that either spatial or temporal coordination is more affected and should be prioritized during rehabilitation.

### 4.3. Neural Mechanisms of Adaptation in PwMS

One surprising finding was that the step length of the affected limb following split-belt treadmill adaptation was unchanged at the group level, while SLA had robust improvements in the responders group. Because our asymmetry measurement was affected step length minus less affected step length, PwMS have different spatial adaptation strategies to successfully account for the belts going different speeds, rather than simply increasing the step length of the leg under the fast belt. This indicates that there is likely more nuanced supraspinal control involved during error correction of stepping that contributes to feedforward adaptation for PwMS.

Split-belt treadmill adaptation relies on both reactive feedback control and predictive feedforward modulation. It has been demonstrated through lesion studies [38] that immediate gait parameter adjustments use feedback mechanisms that are not dependent on supraspinal control, with modulation happening at the spinal cord level, likely through central pattern generators. To maintain the effects of split-belt treadmill adaptation and experience feedforward aftereffects, cerebellar influence is essential [39]. In the cerebellum, the midline vermis and fastigial nuclei have been suggested to modulate posture and locomotion [40]. Along with this, Purkinje cell firing rates have been shown to increase robustly during split-belt treadmill walking in decerebrate cats which demonstrates the involvement of the cerebellum during locomotor adaptation [41]. For successful phasic bilateral coordination, the cerebrum, specifically the sensorimotor integration areas, coordinate various afferent information to optimize motor output for successful motor planning. The interaction between the cerebellum and cerebrum is also necessary for adaptation, with studies finding that cerebellar brain inhibition is proportional to motor learning outcomes [42]. This observation that spatial and temporal adaptations are independent suggests that there may be different neural mechanisms for spatial and temporal modulation, which is in line with other findings [32]. Choi and colleagues found that in individuals with hemispherectomy, their spatial feedforward adaptability was unaffected while their temporal feedforward adaptability was significantly impaired, which implies that temporal coordination may rely more heavily on the cerebrum while spatial coordination may rely more heavily on the cerebellum [43].

Communication of afferent information, primarily proprioception, during gait via the dorsal spinocerebellar tract and the posterior column–medial lemniscus is essential for accurate motor output and, therefore, necessary for successfully updating motor programs. It has been recently established that while PwMS have motor impairments, it is likely the mechanism of these motor impairments originates from inaccurate sensory signaling due to damaged myelin in the spinal cord and brain regions that coordinate bilateral movements [22]. The majority of split-belt treadmill adaptation rehabilitation research focuses on stroke or Parkinson’s disease, where gray matter or cerebral injury likely causes impaired motor output. While the cerebellum is the primary region for sensorimotor adaptation [44], due to the pathology of MS, there may be a unique combination of cerebral and cerebellar contributions to motor learning to compensate for poor spinocerebellar signaling [34], such as enhanced cerebral influence. 

### 4.4. Future Directions

With the unique pathology of sensory impairment in MS, further investigation of neural mechanisms of feedforward adaptation in this population is warranted. Future phases of this study include coupling split-belt treadmill adaptation with functional near-infrared spectroscopy (fNIRS) to measure cortical activation during sensorimotor adaptation compared to neurotypical peers [45]. While split-belt treadmill adaptation has shown acute decreases in gait asymmetry, few studies have successfully implemented this paradigm to improve gait coordination in the long term. Some studies have combined multiple sessions of split-belt treadmill training with cerebellar stimulation with varying success [46,47]. For PwMS, the impairment that leads to gait asymmetry likely lies in sensorimotor white matter transduction [36]; thus, amplifying sensory signaling may be beneficial to further enhance feedforward storage of new walking patterns. One mechanism that has successfully improved sensory signaling is transcutaneous electrical nerve stimulation (TENS) in the periphery, which has been shown to increase excitability of peripheral afferent neurons and consequently improve motor coordination [48]. Pairing lower limb TENS with split-belt treadmill adaptation coupled with fNIRS allows for assessment of the influence of amplified sensory signaling on cortical activation and allows for investigation of amplified sensory signaling to improve feedforward sensorimotor adaptation. This would further inform clinicians on the utility of adaptation paradigms, such as split-belt treadmill training, coupled with enhanced afferent signaling as long-term rehabilitation strategies for PwMS.

### 4.5. Limitations

PwMS are known to have cognitive deficits, and split-belt treadmill adaptation has been shown to increase cognitive load [49]. Another predictor of responders and nonresponders may have been visuospatial cognition, as shown in recent evidence [50]. Our cognitive test (MOCA) was not sensitive enough to detect this, and an assessment such as the Trail Making Test could add an important distinction between participants, potentially being a predictor for the magnitude of aftereffects following split-belt treadmill adaptation. Another limitation was that this sample of PwMS was relatively healthy, with the majority of participants having a lower disability level than typical when considering years since diagnosis. If this sample was more representative of the MS population, there may have been a greater number of responders to split-belt treadmill adaptation. However, an inherent limitation of split-belt treadmill adaptation is the required capability to complete the demanding walking task, which unfortunately excludes many individuals with a higher disability level and limits generalizability. Another beneficial addition to this study would be a longer tied-belt walking session following split-belt treadmill adaptation. This would allow for the recording of the rate of de-adaptation following the aftereffects and be informative of group differences during de-adaptation.

## 5. Conclusions

This is one of the first studies to investigate sensorimotor adaptability of locomotion in PwMS, a population that has pronounced sensory impairments [36]. These data demonstrated that PwMS maintain the ability to adapt their gait cycle parameters to improve symmetry, with those experiencing the greatest asymmetry at baseline having the greatest magnitude of symmetry improvements following split-belt treadmill adaptation. Additionally, spatial and temporal gait cycle parameters adapted independently, suggesting that there are separate neural mechanisms for feedforward adjustment of these parameters. Further investigation into enhancing neuroplastic change and understanding the neural mechanisms of adaptation in PwMS will be informative to rehabilitation strategies to further improve locomotion in PwMS and the many other populations who experience gait dysfunction.

## Figures and Tables

**Figure 1 sensors-23-05456-f001:**
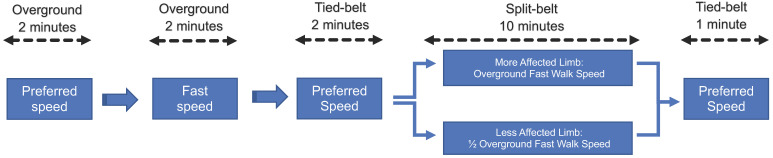
Participants completed 5 independent walking trials including baseline overground walking, baseline tied-belt treadmill walking, split-belt treadmill walking, and post-adaptation tied-belt walking.

**Figure 2 sensors-23-05456-f002:**
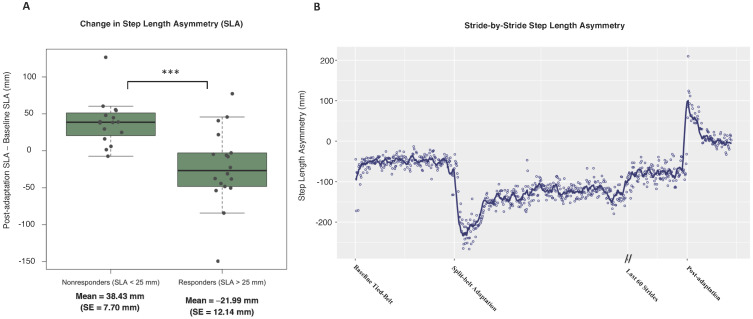
(**A**) Change in step length asymmetry (SLA) from baseline to post split-belt treadmill adaptation. It was hypothesized that participants with a low baseline SLA (better spatial symmetry) would experience minimal symmetry improvement following split-belt treadmill adaptation. Here, participants are grouped based on their baseline SLA as above or below the total sample median of 25.48 mm, with a significant difference between responders and nonresponders (*p* < 0.001, d = 1.40). (**B**) Stride-by-stride representation of SLA for a single participant. At baseline, the step length of the more affected limb is less than the step length of the less affected limb (spatial asymmetry). Following split-belt adaptation, the aftereffects demonstrate an improvement in spatial symmetry represented by a smaller SLA. • = individual participant change, *** = *p* < 0.001.

**Figure 3 sensors-23-05456-f003:**
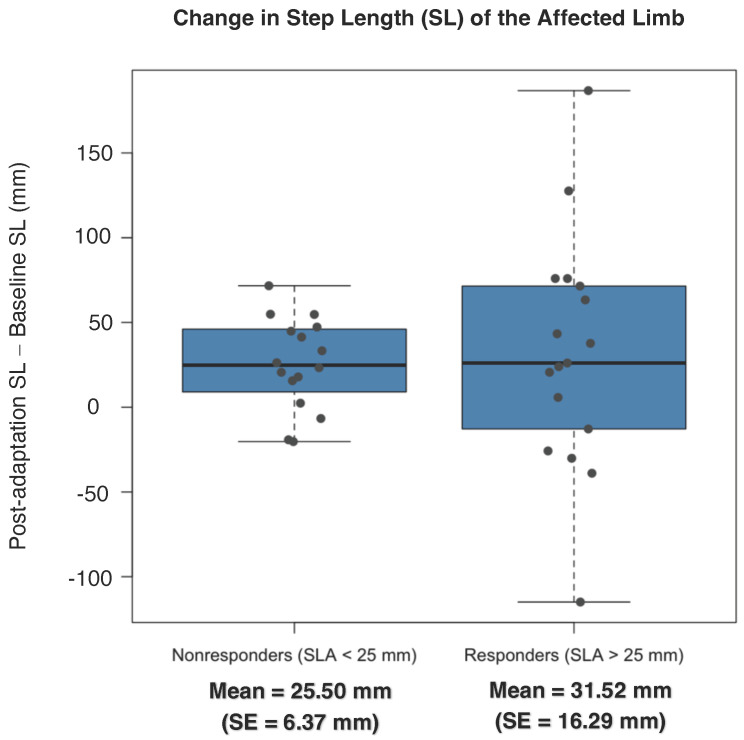
Change in step length of the affected limb from baseline to post split-belt treadmill adaptation. Participants are grouped using the same method from Figure 2. Contrary to the hypothesis, there was no change in step length of the affected limb following split-belt adaptation (*p* = 0.74, d = −0.11). • = individual participant change.

**Figure 4 sensors-23-05456-f004:**
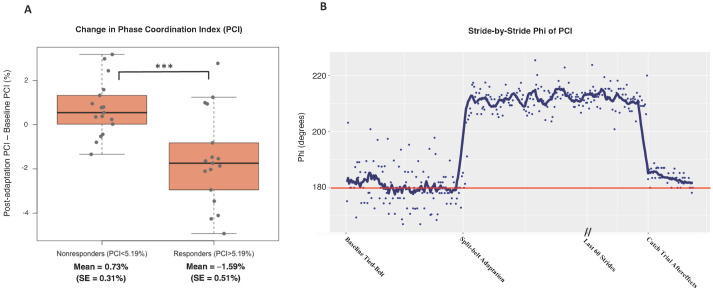
(**A**) Change in PCI from baseline to post split-belt treadmill adaptation. It was hypothesized that participants with a low baseline PCI (better temporal symmetry) would experience minimal symmetry improvement following split-belt treadmill adaptation. Here, participants are grouped based on their baseline PCI value as above or below the total sample median of 5.19% with a significant difference between responders and nonresponders (*p* < 0.001, d = 1.33). (**B**) Stride-by-stride representation of the phi component of PCI for a single participant with 180°, or the red line, depicting perfect temporal symmetry between limbs. At baseline, phi has large variability around 180°. During split-belt adaptation, phi is increased, due to forced temporal misalignment. Following, the aftereffects demonstrate less variation in phi, which indicated more ideal temporal coordination. • = individual participant change, *** = *p* < 0.001.

**Figure 5 sensors-23-05456-f005:**
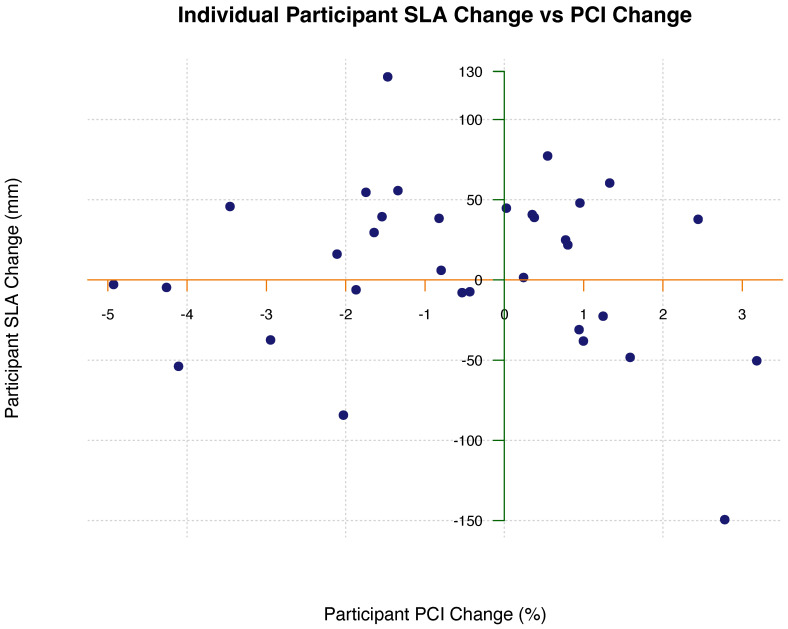
Change in step length asymmetry (SLA) plotted against change in phase coordination index (PCI). Pearson’s correlation coefficient showed no correlation between SLA change and PCI change (r = −0.12, *p* = 0.49), suggesting that spatial and temporal gait parameters adapt independently of each other. • = individual participant change.

**Table 1 sensors-23-05456-t001:** Participant Characteristics. Reported mean and standard deviation (SD) of selected demographics, symptoms, and test scores. Overall, this cohort of PwMS was quite active and had mild symptoms compared to other cohorts with similar years since diagnosis.

Participant Characteristic	Mean	SD
N	35	
Age	51.66	12.02
Sex	61% Female	
BMI	25.37	4.19
Activity (min per week)	289.4	266.8
Years since diagnosis	13.85	10.73
Falls in last 6 months	0.65	1.02
Reported neuropathy	85%	
EDSS	3.57	1.03
MFIS	31.39	14.98
MSWS-12	21.89	12.08
BDI	7.63	7.06
MOCA	27.25	2.30

## Data Availability

The data presented in this study are available on request from the corresponding author. The data are not publicly available due to participant privacy.

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
