# Peer review of "Split-Belt Treadmill Adaptation Improves Spatial and Temporal Gait Symmetry in People with Multiple Sclerosis"

_sensors, 2023, doi:10.3390/s23125456_

Round 1

Reviewer 1 Report

In the current study, Hagen et al. assessed the advantages of split-belt treadmill adaptation to decrease gait asymmetries in people with multiple sclerosis by using the Step length asymmetry and phase coordination indexes. Authors have included 35 subjects with multiple sclerosis for the assessment. Accordingly, multiple sclerosis patients with worse baseline symmetry exhibits higher response to split-belt treadmill adaptation along with a significant difference between predicted responders and nonresponders for subjects with improved gait symmetry. Overall manuscript is constructed well with detailed statistical presentation in the methods section with occasional grammatical errors present throughout the manuscript. However, the study has major limitations. A detailed clinical neurologic and motor assessment of the recruited subjects could have given deep insights into the extent of multiple sclerosis. As authors have pointed out that study subjects were relatively healthy with a lower disability level and the results may not be applicable to subjects with multiple sclerosis with higher disability level.

Occasional grammatical errors present throughout the manuscript.

Reviewer 2 Report

The article entitled “Split-belt Treadmill Adaptation Improves Spatial and Temporal Gait Symmetry in People with Multiple Sclerosis”. The aim was to

determine if spatial and temporal gait parameters are adaptable in PwMS during split-belt treadmill adaptation and if this adaptation can lead to a decrease in gait asymmetry.

Below are some suggestions:

In the Abstract:

- The abstract is well described, but I suggest that the authors make the purpose of the study more clear.

1. In the Introduction:

- The introduction is objective, but superficial, I suggest that the authors deepen the importance of this treatment strategy and its clinical relevance.

2. Materials and Methods:

- Authors must enter Ethics Committee approval.

- The methodology is well described, but I suggest an clinical design at the beginning, summarizing all the steps carried out in the research.

3. Results 

- I suggest improving the quality of figure 2 (A), 3, figure 4 (A) for better viewing. 

4. Discussion 

I suggest the following modifications to the discussion: 

- start the first paragraph with a summary of what was performed and main results, positioning with the current world. 

5. Conclusion 

- I suggest inserting a conclusion with final considerations, including the purpose of the research.

** Due to the importance of this journal, authors should insert and cite a greater number of references throughout the article** A better scientific basis is needed.

Moderate editing of English language required.

Reviewer 3 Report

I would like to congratulate the authors of the present manuscript as it well presents the results of an interesting study involving the rehabilitation and training of gait in people with MS, and with particular attention to the effect of asymmetries.

The paper is well written, the methods clear and the results well presented and discussed. I would just add a question if the authors have considered how potential asymmetries might play a role in both biomechanical and physiological characteristics, as discussed for example in terms of the energy cost of locomotion that could play a role in the development of fatigue (see as an example: doi: 10.1007/s00421-019-04295-3)

Round 2

Reviewer 1 Report

Authors have addressed the comments.

Occasional grammatical errors are present throughout the manuscript.